# In-situ study of the impact of temperature and architecture on the interfacial structure of microgels

Steffen Bochenek[1], Fabrizio Camerin[2,3], Emanuela Zaccarelli [2,3], Armando Maestro [4,5,6], Maximilian M. Schmidt[1], Walter Richtering[1] & Andrea Scotti [1✉]

The structural characterization of microgels at interfaces is fundamental to understand both their 2D phase behavior and their role as stabilizers that enable emulsions to be broken on demand. However, this characterization is usually limited by available experimental techniques, which do not allow a direct investigation at interfaces. To overcome this difficulty, here we employ neutron reflectometry, which allows us to probe the structure and responsiveness of the microgels in-situ at the air-water interface. We investigate two types of microgels with different cross-link density, thus having different softness and deformability, both below and above their volume phase transition temperature, by combining experiments with computer simulations of in silico synthesized microgels. We find that temperature only affects the portion of microgels in water, while the strongest effect of the microgels softness is observed in their ability to protrude into the air. In particular, standard microgels have an apparent contact angle of few degrees, while ultra-low cross-linked microgels form a flat polymeric layer with zero contact angle. Altogether, this study provides an in-depth microscopic description of how different microgel architectures affect their arrangements at interfaces, and will be the foundation for a better understanding of their phase behavior and assembly.

[1] Institute of Physical Chemistry, RWTH Aachen University, Landoltweg 2, 52074 Aachen, Germany. [2] CNR-ISC, Sapienza University of Rome, Piazzale Aldo Moro 2, 00185 Roma, Italy. [3] Department of Physics, Sapienza University of Rome, Piazzale Aldo Moro 2, 00185 Roma, Italy. [4] Institut Laue-Langevin ILL DS/LSS, 71 Avenue des Martyrs, 38000 Grenoble, France. [5] Centro de Fısica de Materiales (CSIC, UPV/EHU) - Materials Physics Center MPC, Paseo Manuel de Lardizabal 5, 20018 San Sebastián, Spain. [6] IKERBASQUE-Basque Foundation for Science, Plaza Euskadi 5, 48009 Bilbao, Spain. ✉email: andrea.scotti@rwth-aachen.de

Soft nano- and microgels—cross-linked polymer networks swollen in a good solvent—reveal peculiar properties that are different from those of other colloidal systems such as hard nanoparticles, polymers, and surfactants[1–5]. The impact of softness, for instance, emerges when micro- and nanogels adsorb at interfaces: they stretch and deform to maximize the coverage of the interface and minimize the interfacial energy[6–11]. At the same time, they do not completely disassemble but remain individual particles, in contrast to other macromolecules such as block copolymer micelles, which irreversibly change their internal conformation upon adsorption at an interface[12,13].

Nano- and microgels based on poly-*N*-isopropylacrylamide (pNIPAM) have a high interfacial activity[14] and at the same time maintain their thermo-responsiveness once adsorbed to air-[15–17], liquid-[18–21], or solid interfaces[22–25]. They can be used to prepare smart emulsions[18,19,26–28] that can be broken on demand as a function of external stimuli such as temperature and pH[18,19,29–32].

A detailed knowledge of the 3D structure of microgels at an interface is essential to understand fundamental aspects such as their 2D-phase behavior[33–43] or their functionality in emulsion stabilization. While there has been significant progress in studying microgels at solid substrates, in situ experiments at fluid interfaces are still scarce. A powerful technique to obtain experimental insight into the structure and composition of surfaces and/or thin films with the sub-nanometric resolution is specular neutron reflectometry (SNR), which has been employed to study NIPAM-based systems, such as linear polymers and nanogels[44,45].

Recently, Zielińska et al. probed the structure of pNIPAM nanogels (with a diameter smaller than 40 nm) below and at the lower critical solution temperature of pNIPAM of 32 °C[44,46]. They found that nanogels protrude for ≈2 nm in the air phase and form a thick polymeric layer at the interface. After this, two layers of highly solvated pNIPAM were observed. As highlighted in these studies, a key aspect that determines the monolayer structure is represented by the nanogel deformability. More generally, the extent of the microgels' deformation, their final shape, and their phase behavior strongly depend on their softness and internal architecture.

It can be expected that the size and cross-linker density of the microgels strongly influence the structure of the microgel-covered interface and indeed a transition from particle-to-polymer-like behavior has been observed for ultra-soft microgels adsorbed to solid interfaces[39]. Atomic force microscopy (AFM), cryo-scanning electron (cryoSEM) microscopy, and computer simulations show that adsorbed standard microgels, i.e., microgels with a cross-linker content of few mol%, have a core-corona or fried-egg-like shape when dried, where the fuzzy shell of the microgels forms a thin layer at the interface with the more cross-linked core in the center[6,8,33,47,48]. The core-corona structure gives rise to a rich 2D-phase behavior of the microgel monolayer characterized by a solid-to-solid phase transition[33]. In contrast, AFM measurements demonstrate that ultra-soft microgels have a flat and homogeneous pancake-like structure[25]. Furthermore, depending on the monolayer concentration, they can form both flat films and behave as polymers or as a disordered arrangement of particles[39].

In this contribution, we address the following questions: Do microgels protrude into the air, and if so how far? Is it possible to determine a contact angle for microgels? How are these quantities affected by the cross-linking density and by the collapse of the microgels in the water phase? In particular, we employ SNR to determine in situ the structure of microgels along the normal to the interface and compare the resulting polymer fraction profiles with those obtained by computer simulations.

We investigate two different types of microgels. The first one is a standard microgel synthesized with a cross-linker content of 5 mol%. This has an architecture characterized by a more cross-linked core and a gradual decrease of the cross-linking density and the polymer segment density towards the periphery. Finally, dangling chains decorate the outer shells[49]. This architecture is a consequence of the fact that the cross-linker agent reacts faster than the monomer during the precipitation polymerization[50]. We prepared two separate batches, where in one case the isopropyl group of the monomer was deuterated to improve the contrast for neutron reflectometry (NR).

pNIPAM microgels can also be synthesized via precipitation polymerization without the addition of a cross-linker agent[51]. The network is formed by self-cross-linking of NIPAM due to transfer reactions[52]. As with the standard microgels, we use a partially deuterated monomer in which the vinyl group is deuterated[52] to increase and vary the contrast in neutron reflectometry. Given the absence of a cross-linker agent, these ultra-low cross-linked (ULC) microgels are ultra-soft[53,54] and have an almost uniform, albeit very low, internal density of polymer segments[39]. Nonetheless, such particles remain fundamentally different from linear polymers. For instance, in bulk solution, ULC microgels were found to form colloidal crystals in clear contrast to linear or branched chains[54,55]. Furthermore, their behavior can be tuned between that of polymer and one of the colloidal particles depending on the compression of the monolayer[39]. These microgels also differ from linear polymers once adsorbed at a solid interface where their architecture is the one of ultra-soft disks[25].

The differences in internal architecture between standard and ULC microgel affect their compressibility and deformability. For instance, the presence of a more cross-linked and denser core inhibits large compression in bulk[56], whereas the poorly cross-linked network of the ULC microgels is easy to compress in crowded solutions[53,57]. While compressibility is the key aspect of the three-dimensional response of microgels, their deformability is pivotal once they are confined in two dimensions, i.e., onto liquid or solid interfaces.

The analysis of our data shows the effects of the microgel internal architecture on their structure orthogonal to the interface. For both systems, the protrusion in the air and the polymeric layer sitting at the interface are independent of the temperature, *T*. Furthermore, simple geometrical considerations on the density profiles combined with the in-plane microgel radius determined by AFM, allow us to determine the apparent contact angle of the adsorbed microgels. We show that the morphology of ULC microgels is more similar to linear polymers and macromolecules, while standard microgels resemble more closely hard colloids.

## Results

**Microgel structure in bulk solution.** The ratio between the hydrodynamic radius in the swollen and collapsed state—swelling ratio—is a good measurement of the softness of the microgel network: The larger this ratio, the softer the microgel[58–60]. All microgels studied here have a comparable hydrodynamic radius at 20 °C, see Table 1 and Supplementary Fig. 1a. They do however exhibit different swelling ratios, see Supplementary Fig. 1b. For the hydrogenated 5 mol% cross-linked standard pNIPAM microgels, 5 mol% D0, the swelling ratio is 1.76 ± 0.03. For the deuterated pNIPAM microgels synthesized with the same amount of cross-linker—5 mol% D7—the swelling ratio is 2.12 ± 0.04. Finally, the swelling ratio of the deuterated pNIPAM ULC microgels, ULC D3, is 2.56 ± 0.05. This confirms that the ULC microgels are the softest, according to this parameter.

**Table 1 Characteristic lengths of the individual pNIPAM-based microgels below and above their VPTT.**

| Name | $T$ | $R_h$ | $R_{SANS}$ | $R_{SANS,c}$ | $2\sigma_{SANS}$ | $2R_{2D}$ | $2R_{2D,c}$ | $h_{2D}$ |
|---|---|---|---|---|---|---|---|---|
| | (°C) | (nm) | (nm) | (nm) | (nm) | (nm) | (nm) | (nm) |
| 5 mol% D0 | 20 | 150 | 151 | 32 | 119 | 688 | 360 | 21 |
| 5 mol% D0 | 40 | 85 | 72 | 59 | 13 | 651 | 289 | 26 |
| 5 mol% D7 | 20 | 153 | 120 | 33 | 87 | – | – | – |
| 5 mol% D7 | 40 | 72 | 62 | 57 | 5 | – | – | – |
| ULC D3 | 20 | 138 | 134 | 53 | 81 | 733 | – | 3 |
| ULC D3 | 40 | 54 | 56 | 41 | 15 | 689 | – | 4 |

Hydrodynamic radius in water, $R_h$, radius from SANS in $D_2O$, $R_{SANS} = R_{SANS,c} + 2\sigma_{SANS}$ where $R_{SANS,c}$ is the core radius in bulk and $2\sigma_{SANS}$ is the fuzziness of the shell in bulk determined by SANS. $2R_{2D}$ is the interfacial (dry) diameter and $2R_{2D,c}$ is the interfacial (dry) diameter of the core. $h_{2D}$ is the maximum height once adsorbed (dry). The last three quantities are determined by AFM, see Supplementary Figs. 3 and 4. The values including the errors are given in Supplementary Table 1.

Small-angle neutron scattering (SANS) is used to determine the characteristic lengths of the microgels, such as the total radius, $R_{SANS}$, the radius of the more cross-linked core, $R_{SANS,c}$, and the extension of the fuzzy shell, $2\sigma_{SANS}$. The values of these quantities are determined by fitting the form factors with the fuzzy-sphere model[49] and are reported in Table 1. The data and the fits in Supplementary Fig. 2a–d confirm the different internal architecture between standard and ULC microgels.

We note that the main effect of selective deuteration and of using deuterated solvents is to shift the VPTT of pNIPAM to a higher temperature[61–64]. However, at the lowest and highest temperatures measured, the microgels are in the fully swollen and collapsed state (see Supplementary Figs. 1c and 2a–d), respectively, allowing for an appropriate comparison of the different architectures.

**Standard microgels at the interface.** For each monolayer of hydrogenated and deuterated microgels studied here, the intensities of the reflected neutrons, R(Q), were recorded as a function of momentum transfer normal to the interface, $Q$, in two isotopic contrasts: $D_2O$ and air contrast matched water (ACMW). The latter consists of a mixture of $D_2O$ and $H_2O$ (8.92% v/v), which matches the scattering length density (SLD) of air ($b_{air} = 0 \times 10^{-4}$ nm$^{-2}$), and therefore only the polymer contributes to the reflected signal of the curves in Fig. 1a, b. The reflectivity curves for the same microgels, measured in $D_2O$ as sub-phase, are plotted in the insets of Fig. 1a, b. In this case, when a neutron beam is reflected from air at $D_2O$, which has a higher SLD ($b_{D_2O} = 6.36 \times 10^{-4}$ nm$^{-2}$) or a lower refractive index $n = 1 - \lambda^2/2\pi b$ (with $\lambda$ the neutron wavelength), respectively, total reflection occurs below a critical value of the momentum transfer $Q_c = 0.16$ nm$^{-1}$. Above this value, the reflectivity decays as a function of $Q^4$.

The samples studied here yielded laterally homogeneous interfaces on the length scale of the in-plane neutron coherence length, on the order of several microns[65]. This implies that the measured SNR can be correlated with the averaged SLD depth profile across the interface delimited by this coherence length and, therefore, the in situ structure of the microgels as a function of the distance from the interface $z$ can be determined. This is done by fitting the reflectivity curves with a model consisting of different layers characterized by a thickness, $d$, a roughness, $\sigma$, and a SLD, $b$. The latter contains information on the atomic density of the NIPAM molecules and, therefore, is linked to the polymer concentration and solvation of the different layers (see Methods section for further details). Here, we find that a model composed of four layers is the most suitable to describe the density profile of the standard pNIPAM microgels perpendicular to the plane of the interface where the layers 1-to-4 are sandwiched between the bulk air (layer 0) and the bulk solvent (background layer).

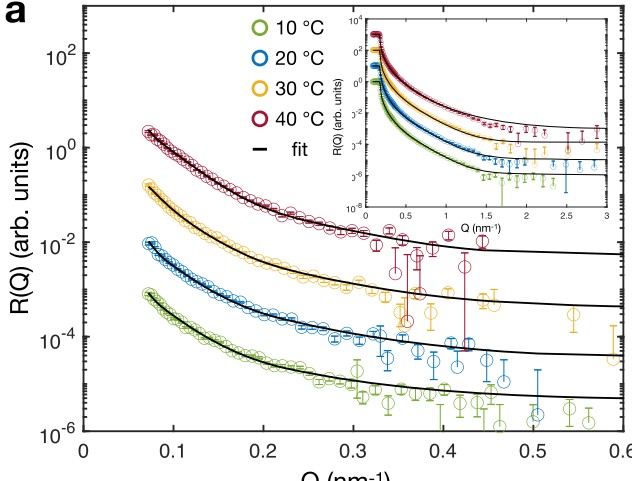

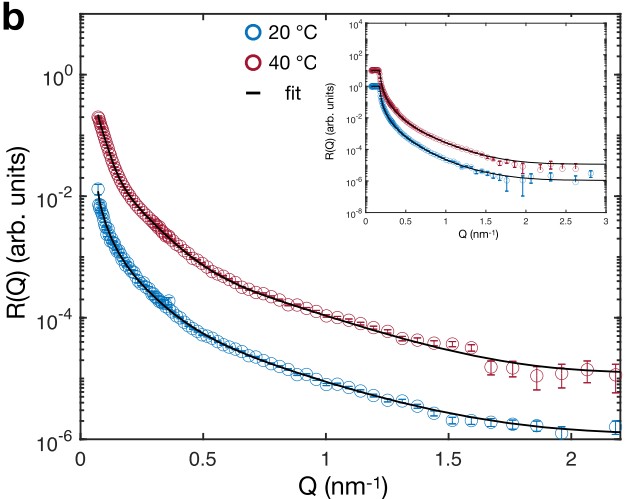

**Fig. 1 Reflectivity curves of 5 mol% cross-linked microgels at different temperatures. a** Reflectivity, R(Q), versus momentum transfer, $Q$, of pNIPAM microgels at the air-ACMW interface and corresponding fits. **b** Reflectivity curves of D7-NIPAM microgels at the air-ACMW interface with fits. Insets: Reflectivity curves at air-$D_2O$ interfaces. The curves are shifted in y-direction for clarity. The unshifted curves are shown in Supplementary Fig. 6a, b. The error bars represent the statistical errors on R(Q).

The length and width of the slabs are delimited by the illuminated area which is roughly $10^9$ times the interfacial diameter of the measured microgel. Therefore, in contrast to microscopy-based techniques, our measurements probe a statistically significant ensemble of microgels. We fit the R(Q)-curves of the same sample at the same temperature for both contrasts simultaneously to reduce the number of free parameters. The best fits are shown by the black full lines in Fig. 1a, b. The parameters of the fits are reported in Table 2. The use of models with a smaller number of layers cannot reproduce the experimental data or it leads to a density profile inconsistent with previous studies[6,10,35,66–69], see Supplementary Fig. 7a–c.

In addition, to verify the validity of the four slab models, the data for the deuterated microgels at 20 °C have been fitted using a continuous variation of the SLD profile sliced into many (>1000) thin layers of 1.5 Å thickness. As shown in the Supporting Information (Supplementary Fig. 10a, b), the fit leads to identical results and, therefore, it confirms the validity of our model. From this discussion, it is clear that the model employed here can

**Table 2 Parameters of the 4-layers fit for the 5% mol% cross-linked microgels in Fig. 1.**

| T | Layer 1 | | | Layer 2 | | | Layer 3 | | | Layer 4 | | | Background | |
|---|---|---|---|---|---|---|---|---|---|---|---|---|---|---|
| | $d_1$ | $\sigma_1$ | $b_1$ | $d_2$ | $\sigma_2$ | $b_2$ | $d_3$ | $\sigma_3$ | $b_3$ | $d_4$ | $\sigma_4$ | $b_4$ | $\sigma_{bkg}$ | $d_{total}$ |
| (°C) | (nm) | (nm) | ($10^{-6}$ Å$^{-2}$) | (nm) | (nm) | ($10^{-6}$ Å$^{-2}$) | (nm) | (nm) | ($10^{-6}$ Å$^{-2}$) | (nm) | (nm) | ($10^{-6}$ Å$^{-2}$) | (nm) | (nm) |
| 5 mol% D0 microgels, $b_{theo} = 0.93 \times 10^{-6}$ Å$^{-2}$ | | | | | | | | | | | | | | |
| 10 | 14 | 8 | 0.06 | 2.1 | 0.7 | 0.32 | 4.4 | 0.4 | 0.14 | 122 | 3.5 | 0.06 | 31 | 220 |
| 20 | 14 | 8 | 0.06 | 2.1 | 0.7 | 0.31 | 4.3 | 0.8 | 0.19 | 117 | 3.5 | 0.07 | 28 | 210 |
| 30 | 14 | 8 | 0.08 | 2.2 | 1.0 | 0.35 | 4.7 | 0.6 | 0.20 | 99 | 4.0 | 0.08 | 29 | 194 |
| 40 | 14 | 7 | 0.10 | 2.7 | 0.5 | 0.35 | 6.8 | 1.0 | 0.23 | 48 | 3.2 | 0.10 | 26 | 140 |
| 5 mol% D7 microgels, $b_{theo} = 4.78 \times 10^{-6}$ Å$^{-2}$ | | | | | | | | | | | | | | |
| 20 | 16 | 11 | 0.1 | 2.3 | 0.5 | 1.58 | 3.0 | 0.2 | 0.49 | 136 | 3.4 | 0.21 | 33 | 245 |
| 40 | 16 | 8 | 0.2 | 2.6 | 0.2 | 1.73 | 4.7 | 0.3 | 0.62 | 66 | 2.6 | 0.26 | 27 | 160 |

$d_i$ is the thickness of a layer with the scattering length density $b_i$. $\sigma_i$ is the roughness between a layer and the layer above it. $d_{total}$ is the total film thickness and $\sigma_{bkg}$ is the roughness between the last layer and the background. The uncertainties from the fits are given as errors in Supplementary Table 2.

reproduce the data with due accuracy and the lowest number of free fitting parameters.

Figure 2a, b shows the polymer fraction normal to the interface (z-distance) of the hydrogenated (5 mol% D0) and deuterated (5 mol% D7) microgels, respectively. These curves are calculated from the SLD profiles obtained from the fits and shown in Supplementary Fig. 8a, b.

We note that the extension of the dangling, highly hydrated polymeric chains at the end of the swollen microgels is accounted for considering the roughness between the last layer and the background, i.e., equals $2\sigma_{bkg}$. The profiles of the polymer fraction normal to the interface show that the microgels deswell in the vertical direction with increasing temperature. The total film thickness $d_{total} = d_1 + \ldots + d_N + 2\sigma_1 + 2\sigma_{bkg}$ is reported in the last column of Table 2.

Below the VPTT, the 5 mol% D0 microgels are fully swollen and have a $d_{total}$ in between $210 \pm 6$ and $220 \pm 5$ nm. Once the microgels are collapsed at 40 °C, they are deswollen and have a thickness of $d_{total} = (140 \pm 5)$ nm. In the literature, a very similar value of the thickness was measured for the same microgels in the swollen and collapsed state with ellipsometry[34]. Also, the deuterated microgels show the deswelling with temperature. The thickness of the monolayer in the swollen and the deswollen state is $d_{total} = 245 \pm 14$ nm and $d_{total} = 160 \pm 2$ nm, respectively; see Table 2.

In our model, the protrusion of the microgel into the air is $d_p = d_1 + 2\sigma_1$ and is calculated using the values given in Table 2. For clarity, we have shifted the position of the polymer profiles along the z-distance to have this protrusion layer at negative distances from the interface, Fig. 2a, b. The unshifted polymer fraction profiles are shown in the Supporting Information, Supplementary Fig. 9a, b.

At 20 °C, the 5 mol% D0 and 5 mol% D7 microgels protrude by $30 \pm 2$ and $37 \pm 2$ nm into the air, respectively. This corresponds to about 10% of the diameter of the swollen microgels in solution or 15% of their $d_{total}$. The protrusion into the air phase does not change significantly with increasing temperature. Geisel et al. determined a protrusion height below 70 nm for microgels of similar size. They noted that this value is the maximum protrusion height according to geometrical calculations from the cryoSEM images and has to be interpreted as an upper limit[6].

The estimated values of $d_p$ allow us to calculate the apparent contact angles of the microgels assuming a simple orthogonal triangle. To this aim, we make use of the total interfacial diameter $2R_{2D}$ of the individual microgels determined by AFM measurements, see Table 1. The apparent contact angle, $\theta_{C,app} = \arctan(d_p/R_{2D})$ is found to be approximately 5° at 20 and 40 °C. Since the corona of the microgels is expected to form a flat layer within the interfacial plane, the interfacial diameter of the core,

$2R_{2D,c}$, can be used instead. This results in $\theta_{C,app} \simeq 9°$ and 11° at 20 and 40 °C, respectively.

The second region is a thin, polymer-rich layer lying at $z = 0$ (Fig. 2a, b). In our model, this region is described by Layer 2 in Table 2. We assume slabs parallel to the interface and, therefore, we only determine an average SLD which is proportional to the average polymer fraction at the interface. Similarly to the protrusion of the microgels in air, also this polymer-rich layer is temperature independent and has a constant volume fraction of $\approx 0.33$, as indicated by the constant values of SLD reported in Table 2. The high polymer content in these regions implies that the network expelled a significant amount of solvent compared to the solvated part in water. Therefore, we can compare the thickness of these two layers ($\approx 40$ nm) to the length of the collapsed shell at high temperatures in bulk, see Table 1, which is found to be much smaller than the thickness of the layers. From this, we can infer that also a part of the more cross-linked core protrudes into the air, as shown in Fig. 2 and in the sketch in Fig. 3a–c.

Our model also reproduces the portion of a microgel in the aqueous phase, i.e., the third region, as shown by the polymer fractions at $z > 0$ in Fig. 2a, b. This portion of the microgel is described by the third and fourth layers, and the corresponding parameters are reported in Table 2. Its extension is calculated as $d_{water} = d_3 + d_4 + 2\sigma_{bkg}$ and shows the strongest reaction to a change in temperature. For the hydrogenated microgels, $d_{water}$ decreases from $178 \pm 5$ to $106 \pm 5$ nm when the temperature increases from 20 to 40 °C. A change in $d_{water}$ from $205 \pm 6$ to $125 \pm 2$ nm for the same temperature increase is determined for the 5 mol% D7 microgels. This collapse is accompanied by an increase of the polymer fraction in layers 3 and 4 for both microgels as indicated by the increases in the values of $b_i$. We note that both below and above the VPTT, the values of $d_{water}$ are smaller than the hydrodynamic diameters of the swollen and collapsed microgels in bulk, $2R_h$ in Table 1. This observation, combined with the large values of the interfacial diameters, indicates a strong deformation of the adsorbed microgels, see Fig. 3a–c. On the other hand, the swelling ratio in 2D, defined as the ratio between $d_{water}$ at 20 and 40 °C, is found to be $1.68 \pm 0.09$ and $1.65 \pm 0.05$ for the hydrogenated and deuterated 5 mol% cross-linked microgels, respectively. These values are smaller than the corresponding ratios in 3D, implying that the adsorption leads to a stiffening of the polymeric networks swollen in water, as also found in computer simulations[37]. Furthermore, provided both microgels have the same 2D swelling ratio, the 5 mol% cross-linked standard microgels have similar softness at the interface, whereas in bulk the deuterated ones appear to be slightly softer.

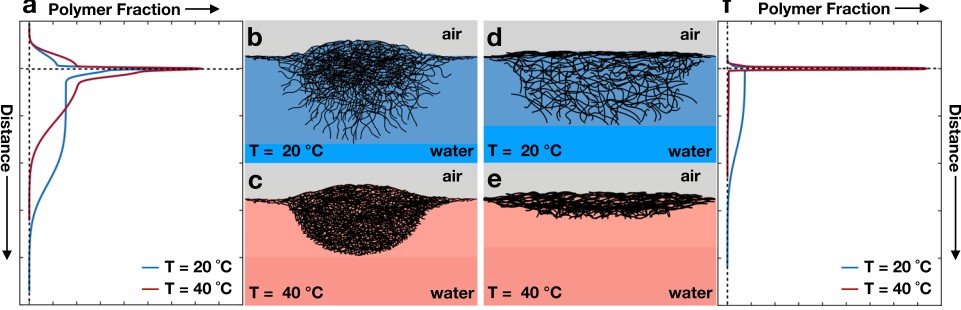

**Fig. 2 Structure of 5 mol% cross-linked microgels at liquid interfaces.** Polymer fractions of the adsorbed 5 mol% D0 (**a**) and 5 mol% D7 (**b**) microgels at different temperatures. **c** Density profiles of simulated microgels at different effective temperatures, corresponding to $\alpha = 0, 0.5$. Horizontal and vertical dashed lines are guides to the eyes and represent zero polymer fraction/density and zero z-distance from the interface, respectively. Negative values of z represent the air phase and positive values represent the water phase. **d** Simulation snapshots showing the side perspective of an adsorbed standard microgel for $\alpha = 0$, and $0.5$. Solvent particles are not shown for visual clarity.

**Fig. 3 Sketch of the adsorbed microgels. a** The vertical profiles of standard microgels and **f** the vertical profiles of ULC microgels below and above the VPTT. Their corresponding shapes are outlined in **b**–**e**. The shapes are based on the combination of our polymer fraction profiles, simulations and AFM measurements at the liquid-solid interface from the literature[25].

We also note that the slight difference in polymer fraction in the water phase between deuterated and hydrogenated nanogels depends on the fact that they have slightly different masses and molecular weights $M_w$. Combining viscosimetry measurements and dynamic light scattering measurements[70], we found that the 5

mol% D7 microgels have a mass of $6.3 \pm 0.6 \times 10^{-19}$ kg ($M_w = 3.8 \pm 0.4 \times 10^8$ gmol$^{-1}$), while the 5 mol% D0 microgels have a mass of $7.7 \pm 0.7 \times 10^{-19}$ kg ($M_w = 4.6 \pm 0.4 \times 10^8$ gmol$^{-1}$).

The conformation of the regular microgel at the interface is in excellent agreement with numerical simulations. In this case,

microgels are synthesized in silico through the self-assembly of patchy particles[37,71]. The resulting polymer network is disordered and accounts for a higher concentration of cross-linkers in the core of the particle, with a bulk density profile that progressively rarefies in the outer corona. The microgel is embedded within two different types of immiscible solvents, mimicking air and water. In this way, the simulated microgel spontaneously acquires the typical fried-egg-like shape. More details on the assembly process and on the simulations at the interface can be found in the Methods section.

In order to compare with the experimental profiles of the microgel parallel to the plane of the interface, we calculate the numerical number density profiles by dividing the simulation box into three-dimensional slabs along the z-direction, i.e., orthogonally to the interfacial plane. In this way, we have direct access to the polymer network without any interference given by the presence of the solvent. The resulting profiles are reported in Fig. 2c for two different effective temperatures.

The three regions described experimentally are also present in the numerical profiles. At all temperatures, we detect the presence of a protrusion into the air phase and a polymer layer lying on the interface. As shown by the snapshots reported in Fig. 2d, the protrusion is given by the fact that the more cross-linked core cannot fully expand, as it happens for the corona, on the interfacial plane. In fact, the corona creates the second part of the density profile that is characterized by a pronounced peak in the interfacial density profile. The polymer network accumulates onto the interface to reduce the surface tension between the two fluids as much as possible. The third region of the profile is inside the aqueous phase. As in the experiments, this region is largely affected by temperature changes. While at low temperatures a large portion of the microgel protrudes significantly into the aqueous phase, at high temperatures the microgel tends to assume a more spherical and compact shape, contracting the polymer chains toward the interfacial plane. The consistency between simulations and experiments also allows us to confirm the robustness of the four layers fitting model used in experiments.

**ULC microgels at the interface**. The reflectivity curves of deuterated ULC microgels at the air-ACMW interface are shown in Fig. 4. In the inset, the measurements with pure $D_2O$ as sub-phase are shown. In contrast to standard microgels, a three-layer model can successfully fit the data (solid lines in Fig. 4). The fit parameters are obtained by fitting reflectivity curves of the same sample at the same temperature simultaneously for both contrasts. The values of the fitting parameters are reported in Table 3.

Once more, we checked the validity of the three-layer model by comparing the results from a fit with a model consisting of a continuous variation of the SLD with many thin layers. In the Supplementary Information, it is shown that the results from the two models are identical (Supplementary Fig. 10c, d). This further demonstrates that a slab model including a Gaussian error function can successfully reproduce the experimental NR data of ULC microgels with the smallest number of free parameters.

The structure of the deuterated ULC microgels as a function of the distance to the interface is described by the shifted and unshifted polymer fraction profiles in Fig. 5a and Supplementary Fig. 9c, respectively. At 20 °C, the length of the protrusion of ULC microgels into air is $d_p = 8 \pm 3$ nm. This is less than 3% of the ULC swollen diameter in solution and approximately 5% of the total thickness of the ULC, $d_{total} = 157 \pm 7$ nm, see Table 3. Analogously to standard microgels, the ULC protrusion into air does not change once the temperature rises above the VPTT. Another similarity with the standard microgels is the presence of a dense layer of polymer sitting sitting at the interface with a

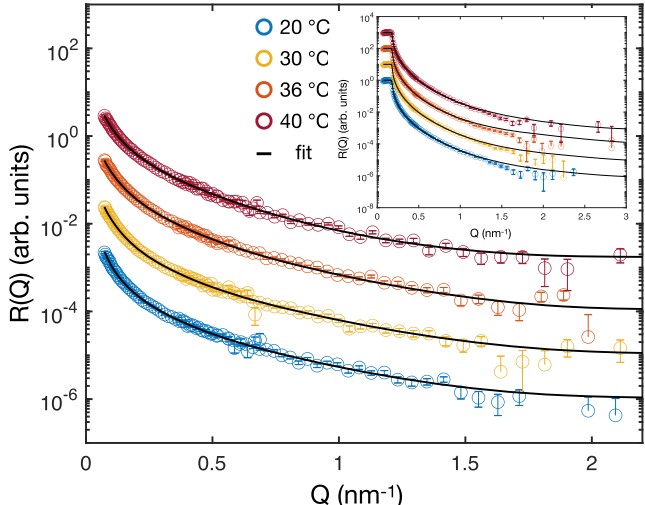

**Fig. 4 Reflectivity curves of ULC D3 microgels at different temperatures.** Reflectivity, R(Q), versus momentum transfer, Q, at the air-ACMW interface. The fits are shown by continuous lines. Inset: Reflectivity curves at air-$D_2O$ interfaces. The curves are shifted in y-direction for clarity. The unshifted curves are shown in Supplementary Fig. 6c. The error bars represent the statistical errors on R(Q).

thickness of ≈ 3 nm. Adding the length of the protrusion in air, $d_p$, to this extension, we obtain ≈11–15 nm which is consistent with the extension of the collapsed fuzzy shell measured by SANS for the D3-ULC microgels, see Table 1. This indicates that, in contrast to standard microgels, only the collapsed external shell protrudes into air and lies on the interface, as shown in Fig. 5 and sketched in Fig. 3d–f.

The third region of the ULC microgels has a lower polymer fraction (below 0.04, Fig. 5a) compared to the standard microgels (above 0.05 Fig. 2a, b) below the VPTT. Unfortunately, due to the resolution of NR and the fact that we average the SLD over the entire monolayer, it is not possible to finely resolve the structure of the collapsed ULC. Above the VPTT, the polymer fraction in the third region of the collapsed ULC is estimated from the value of $b_3$ to be ≈ 0.003. This small value might result from the average between regions with no polymer and denser globules of collapsed microgels around the few cross-linking points. Such globules have been observed by AFM on re-hydrated ULC microgels adsorbed onto solid interfaces after transferring from a Langmuir-Blodgett trough[38].

As for the regular microgels, we can use the estimated $d_p$ and the 2D radius of the ULC microgels to compute their apparent contact angles. The resulting angles are negligible being ≈1° at both temperatures. This behavior is close to what one would expect for macromolecules adsorbed at interfaces in contrast to colloidal particles. This is consistent with recent literature on these ultra-soft microgels[25,54]. Indeed, it has been shown that, due to their high compressibility and deformability, these microgels show the typical behavior of polymers. For instance, their bulk viscosity does not diverge in the proximity of the glass transition but at much higher concentrations, indicating a high degree of deformability[72]. Also, it has been shown that they can cover the interface uniformly as a linear polymer or create a disordered array of individual particles as hard colloids depending on their concentration[39].

To gain more information on the adsorbed ULC microgels, we also performed computer simulations of such system. The corresponding density profiles and simulation snapshots are reported in Fig. 5b, c, respectively. At both effective temperatures,

**Table 3 Summary of the model fits the reflectivity curves of the ULC D3 microgels in Fig. 4.**

| T | Layer 1 | | | Layer 2 | | | Layer 3 | | | Background | |
|---|---|---|---|---|---|---|---|---|---|---|---|
| | $d_1$ | $\sigma_1$ | $b_1$ | $d_2$ | $\sigma_2$ | $b_2$ | $d_3$ | $\sigma_3$ | $b_3$ | $\sigma_{bkg}$ | $d_{total}$ |
| (°C) | (nm) | (nm) | ($10^{-6}$ Å$^{-2}$) | (nm) | (nm) | ($10^{-6}$ Å$^{-2}$) | (nm) | (nm) | ($10^{-6}$ Å$^{-2}$) | (nm) | (nm) |
| ULC D3 microgels, $b_{theo} = 2.57 \times 10^{-6}$ Å$^{-2}$ | | | | | | | | | | | |
| 20 | 3 | 2 | 0.04 | 2.2 | 0.4 | 1.01 | 86 | 0.4 | 0.09 | 30 | 157 |
| 30 | 3 | 2 | 0.070 | 2.4 | 0.4 | 1.08 | 64 | 0.2 | 0.09 | 26 | 125 |
| 36 | 3 | 2 | 0.110 | 2.6 | 0.2 | 1.08 | 61 | 0.2 | 0.05 | 25 | 120 |
| 40 | 3 | 1 | 0.120 | 2.7 | 0.4 | 1.08 | 52 | 0.4 | 0.008 | 15 | 89 |

$d_i$ is the thickness of a layer with the scattering length density $b_i$. $\sigma_i$ denotes the roughness between a layer and the layer above it. $d_{total}$ the approximated total film thickness and $\sigma_{bkg}$ the roughness between the last layer and the background. The uncertainties from the fits are given as errors in Supplementary Table 3.

**Fig. 5 Structure of ULC microgels at liquid interfaces. a** Results of the fits of the experimental data for the ULC D3 microgels. Inset: Zoom of the polymer fraction profiles. **b** Density profiles of simulated ultra-low cross-linked microgels at different effective temperatures, corresponding to $\alpha = 0$, and 0.5. Horizontal and vertical dashed lines are guides to the eyes and represent zero polymer fraction and zero z-distance from the interface, respectively. Negative values of z represent the air phase and positive values represent the water phase. **c** Simulation snapshots showing the side perspective of an adsorbed ULC microgel for $\alpha = 0, 0.5$. Solvent particles are not shown for visual clarity.

the ULC microgels show a flat profile. The polymer network appears to be equally distributed across the interface, with only a slight preference for the water phase. Consistently with experiments, no effect of temperature change is observed for the fraction of polymer in the air side and on the plane of the interface. Furthermore, as in the experiment, the contact angle for the ULC microgels is virtually zero.

For standard microgels, the presence of a well-defined core generates a noticeable dense protrusion into the aqueous phase (Fig. 2a); for the ULC microgels, the amount of polymer in water is considerably lower (Fig. 5a). The ULC microgels extend into the aqueous phase for $d_{water} = d_4 + 2\sigma_{bkg} = 144 \pm 8$ nm at $T = 20$ °C. Furthermore, they remain thermo-responsive and their extension in water decreases to $d_{water} = 83 \pm 3$ nm when the temperature changes from 20 to 40 °C. The 2D swelling ratio equals $1.7 \pm 0.1$, a value much smaller than the corresponding 3D ratio and comparable to the swelling ratio of the standard microgels in 2D. This implies that also ULC microgels experience a significant stiffening of the polymeric network in water due to their large deformation. This takes place both in the lateral and in the vertical directions, as indicated by their large in-plane diameter and by the fact that $d_{water} \ll 2R_h$, see Table 1. Furthermore, $d_{water}$ at 40 °C is slightly larger than the region with a more homogeneous polymer distribution of the collapsed ULC as measured by SANS, see Table 1. Therefore, we can assume that this region does not protrude into the air as shown in the sketch in Fig. 3d, e.

While the experimental and numerical descriptions of ULC microgels agree regarding the microgel portion which protrudes in air and sits onto the interface, there is a difference in what we

observe in the water phase. This is most likely generated by the presence of a few dangling chains that do not absorb on the plane of the interface and, therefore, protrude into the aqueous phase. The reason why this protrusion is not observed in the numerical profiles is most likely due to the small size of the simulated microgel. In fact, the number of monomers and the minimal percentage of cross-linkers employed for the in silico synthesis cause the microgel to be highly extended allowing for all simulated monomers to absorb at the interface. On the contrary, we expect that a significantly larger microgel would have enough monomers to form a plain layer at the interface so that some chains would be desorbed into the aqueous phase, as is the case in experiments. Nevertheless, at present, this is computationally unfeasible due to the huge number of particles that would be involved in an explicit solvent simulation with such a large-sized microgel. For the same reason, an accurate quantitative comparison between numerical and experimental density profiles is, at the moment, out of reach.

## Discussion
In this article, we used neutron reflectometry and computer simulations to probe the structure of microgels orthogonal to the air-water interface, below and above the VPTT. The advantage of neutron reflectometry is that it allows to probe the structure of a statistically significant ensemble of microgels in situ at the interface. Using SNR, we can directly measure the protrusion of the microgels in the air and estimate how it changes with temperature. Microscopy-based techniques, such as transmission

X-ray microscopy (TXM) or cryoSEM, are usually limited by the small number of observed particles, the size of the particles, an observation direction perpendicular to the interface, and complicated sample preparation[6,8,10,66]. The latter makes it particularly difficult, for example, to observe the effect of temperature on the swelling of microgels.

In the future, super-resolved fluorescence microscopy techniques, which in principle can resolve sizes below 30 nm[5], could also be used at the air-water interface to obtain complementary data. To date, however, even these techniques are limited by the spatial resolution in the z-direction which is ≈ 60 nm[73] and by the difficulties in the analysis of the point clouds generated by the blinking of the dyes[74,75].

For both 5 mol% cross-linked and ultra-low cross-linked microgels, we find that the portion of microgels protruding in the air is insensitive to changes in temperature (Fig. 3a, f). Concerning standard microgels, the more cross-linked core is found to partially protrude in the air, leading to an estimate of the apparent contact angle of a few degrees (Fig. 3b, c). This value is significantly smaller than the angle estimated using cryoSEM and TXM of microgels protruding into different n-alkanes[6,66]. The reason for this discrepancy is probably that the cryoSEM estimates were limited either by the smallest angle employed, which was about 30°[6], or by the size of the employed microgels[66].

In contrast, ULC microgels form a flat polymer layer that protrudes only a few nanometers into the air, resulting in a nearly null apparent contact angle (Fig. 3d, e). We also note that the length of such a layer is approximately equal to the extent of its collapsed fuzzy shell (Table 1), supporting the idea that only this part protrudes into the air. Again, since these microgels are ultra-soft and extremely deformable, they stretch as much as possible after adsorption at the interface to minimize the interfacial energy. This behavior is consistent with the experiments of Richardson and co-workers who used neutron reflectometry to probe linear pNIPAM solutions and nanogels with a mesh-size comparable to their dimensions and, therefore, highly stretchable at the interfaces[44,45]. Above the pNIPAM LCST, the collapsed film protrudes about 4 nm into air[45], which is practically the same as the protrusion height estimated here for the ULC microgels. These observations are consistent with the fact that the adsorbed ULC microgels behave more like linear polymers rather than rigid particles[39].

The present study can also contribute to the current debate on the role and importance of capillary interactions for microgels adsorbed at the interface, which seem to be significant only for large particles[76,77]. Indeed, the strength of capillary interactions depends on the size of the particles, the density difference between the particles and the liquid, and the contact angle[78]. Therefore, our measurements reinforce the idea that for small microgels with low contact angle, such as the one investigated here, capillary forces are negligible. Recent literature has also shown that the substitution between air and alkanes, such as decane, only slightly changes the stretching of the microgels at the interface[36]. This is due to the high interfacial tension of the two systems and the insolubility of the microgels in the alkane/oil. However, at lower interfacial tensions, a greater reduction in the spreading of the microgels is observed[79]. Therefore, we expect that our results on the protrusion of the microgels into the hydrophobic phase and the observed difference between ULC and standard microgels at an alkane/(oil)-water interface will not change qualitatively.

Finally, our work is important to shed light on the collective behavior of microgels at interfaces. The differences we highlighted in the structure may be relevant for a more comprehensive understanding of microgels' effective interactions, paving the way for a better description of their 2D assembly and for a clever design of their applications such as emulsion stabilizers.

## Methods

**Synthesis**. Standard 5 mol% D0 (SFB985_B8_SB_M000325), 5 mol% D7 (SFB985_A3_MB_M000238), and ULC D3 (SFB985_A3_MB_M000301) microgels were synthesized by precipitation polymerization[34,52,56,57]. The main monomers for all microgels were NIPAM (D0) or deuterated NIPAM, in which three (D3) or seven (D7) hydrogen atoms have been exchanged by deuterium. The deuterated monomers were obtained from Polymer Source, Canada, hydrogenated monomers were obtained from Acros Organics, Belgium. Surfactants, sodium dodecyl sulfate (SDS) or cetyltrimethylammonium bromide (CTAB), were added during the synthesis to control the size polydispersity and final microgel size. Briefly, for the three different syntheses, 5.4546 g of D0-NIPAM (5 mol% D0 microgels), or 1.5072 g of D7-NIPAM (5 mol% D7 microgels), or 1.0093 g of D3-NIPAM (ULC D3 microgels) were dissolved in 330, 83, and 70 mL double-distilled water, respectively. For the 5 mol% microgels 0.3398 g (5 mol% D0) or 0.1021 g (5 mol% D7) of the cross-linker N,N'-methylenebisacrylamide (BIS) were added. No additional cross-linker was included during the synthesis of the ULC D3 microgels. The reaction flask of the 5 mol% D0 microgels contained additionally 0.1474 g of N-(3-aminopropyl) methacrylamide hydrochloride (APMH) as co-monomer. The monomer solutions were purged with nitrogen under stirring and heated to 65 °C (5 mol% D0), 70 °C (5 mol% D7), and 70 °C (ULC D3). The initiators and the surfactants were dissolved in a few milliliters of double-distilled water in separated vessels and degassed for at least one hour. For the deuterated 5 mol% D7 and ULC D3 microgels 0.372 g and 0.0506 mg of potassium peroxydisulfate (KPS) and 0.202 g and 0.0277 g of SDS were used, respectively. For the 5 mol% D0 microgels, 0.2253 g 2,2'-Azobis-(2-methyl-propionamidin) dihydrochlorid (V50) and 0.0334 g of CTAB were used. After adding the surfactant to the reaction flask, the polymerization was initiated by injecting the dissolved initiators. The reactions were carried out for 4 h at the given temperatures and under constant nitrogen flow and stirring. The obtained microgels were purified by threefold ultra-centrifugation and re-dispersion in fresh double-distilled water. Lyophilization was applied for the storage of all microgels.

**Dynamic light scattering**. A laser with vacuum wavelength $\lambda_0 = 633$ nm was used to probe diluted suspensions of the different microgels in water and heavy water. The temperature was change from 20 to 50 °C in steps of 2 °C using a thermal bath filled with toluene to match the refractive index of the glass. The momentum transfer $Q = 4\pi/\lambda \sin\theta$, was changed by varying the scattering angle, $\theta$, between 30 and 130 degrees, in steps of 5 degrees.

**Small-angle neutron scattering**. SANS experiments were performed at the KWS-2 instrument operated by the JCNS at the MLZ, Garching, Germany, and at the D11 instrument at the Institut Laue-Langevin (ILL, Grenoble, France). For the KWS-2 experiments, the q-range of interest was covered by using a wavelength for the neutron beam of $\lambda = 0.5$ and 1 nm and three sample-detector distances: 20, 8, and 2 m. The detector is a 2D-$^3$He tubes array with a pixel-size of 0.75 cm and a $\Delta\lambda/\lambda = 10$%. For the D11 three configurations were used: sample detector distance, $d_{SD} = 34$ m with $\lambda = 0.6$ nm; $d_{SD} = 8$ m with $\lambda = 0.6$ nm; and $d_{SD} = 2$ m with $\lambda = 0.6$ nm. Due to the velocity selector, the resolution in $\lambda$ was 9%. The D11 is equipped with a $^3$He detector with a pixel size of 7.5 m.

**Compression isotherms and depositions**. Gradient Langmuir-Blodgett type depositions[33,34,36] from air-water interfaces were performed to study the mechanical properties of the microgels and microgel monolayers and visualize them ex-situ. The Langmuir-Blodgett trough was made from polyoxymethylene (POM) and was equipped with two movable POM barriers. For each deposition, the trough was carefully cleaned, heated to the appropriate temperature (20 or 40 °C) with an external water bath, and a fresh air-water interface was created. The surface pressure was monitored during the depositions with an electric balance fitted with a platinum Wilhelmy plate. The substrates were rectangular pieces of ultra-flat silicon wafer (≈1.1 x 6 cm, P100). The substrates were carefully cleaned with distilled water, isopropyl alcohol, and ultrasonication. They were mounted to the dipper arm of the Langmuir-Blodgett trough with an inclination with respect to the liquid interface of about 25°. After moving the substrate to the starting position, the microgels were spread at the air-water interface. For this purpose, microgels were suspended either in 50/50 vol% mixtures of water-propan-2-ol in water and chloroform. This was done to maximize the adsorption of the microgels to air-water interfaces and minimize partial loss of microgels into the sub-phase. This loss is unavoidable if the surface-active component is soluble in either phase. After equilibration for at least 30 min, the substrates were lifted through the interface while the barriers of the Langmuir-Blodgett trough compressed the interface. The speed of the barriers ($v_{barrier} = 6.48$ cm$^2$ min$^{-1}$) was matched to the speed of the dipper arm ($v_{dipper} = 0.15$ mm min$^{-1}$). This, together with the tilt of the substrate, allowed the microgels to be deposited on the substrate with increasing concentration[33].

**Atomic force microscopy**. Deposited, dried microgels were imaged using a Dimension Icon atomic force microscope with closed loop (Veeco Instruments Inc., USA, Software: Nanoscope 9.4, Bruker Co., USA) in tapping mode. The probes were OTESPA tips with a resonance frequency of 300 kHz, a nominal

spring constant of 26 Nm$^{-1}$ of the cantilever and a nominal tip radius of <7 nm (Opus by Micromasch, Germany).

**Image analysis**. The open-source analysis software Gwyddion 2.54 was used to process the AFM images. All images were leveled to remove the tilt and zero height was fixed as the minimum $z$-value of the image.

Height profiles of single dried microgels were extracted through their apices and at different angles with respect to the fast scan direction. Multiple height profiles of one image were summarized and aligned to the apices (zero coordinate of the $x$-axis) to obtain averaged microgel profiles and not to bias the results. The profiles are presented with standard deviations as the error. The apices and heights of microgels were computed using the Matlab function findpeaks.

The AFM phase images were used to determine the interfacial (dry) diameter, $2R_{2D}$, of all microgels, and the interfacial (dry) diameter of the core, $2R_{2D,c}$, of the standard microgels. For this, the interfacial areas, $A_{2D}$ and $A_{2D,core}$, of at least 200 well separated, isolated, and uncompressed microgels were measured. $2R_{2D}$ and $2R_{2D,c}$ were calculated by $2R_{2D} = \sqrt{(4 \cdot A_{2D})/\pi}$.

**Specular neutron reflectometry**. Specular neutron reflectometry measurements were conducted on FIGARO, a time-of-flight reflectometer at the Institute Laue-Langevin, Grenoble, France. Two angles of incidence ($\theta_{in} = 0.615$ and $3.766°$) and a wavelength resolution of 7% $\Delta\lambda/\lambda$ were used yielding a momentum transfer of $0.089 < Q < 3.5$ nm$^{-1}$, normal to the interface. The wavelength of the neutron beam, $\lambda$, was 0.2–3 nm.

An area of $\approx 10 \times 40$ mm$^2$ was illuminated with the neutron beam. The reflected neutrons were detected by a two-dimensional $^3$He detector. The raw time-of-flight experimental data at these two angles of incidence were calibrated with respect to the incident wavelength distribution and the efficiency of the detector. Using COSMOS[80], in the framework of LAMP[81], yielded the resulting reflectivity profiles R(Q), where $R$ is defined as the ratio of the intensity of the neutrons scattered at the air-water interface over the incident intensity of the neutron beam.

SNR experiments were performed using D$_2$O and 8.92% v/v D$_2$O:H$_2$O mixtures as sub-phase. The latter is generally known as air contrast matched water (ACMW) since its scattering length density is equal to the one of air. A polytetrafluoroethylene (PTFE) Langmuir trough with an area of 100 cm$^2$ and a volume of $\approx 60$ mL equipped with two parallel moving PTFE barriers was used. The trough was placed inside a gas-tight box with heated sapphire or quartz glass windows to prevent condensation. The box is placed on an active anti-vibration stage which can be moved vertically and horizontally. Prior to a measurement series (measurements at different temperatures), the trough was carefully cleaned and a fresh air-water (D$_2$O or ACMW) interface was created. For temperature control, the trough was connected to an external water bath. The trough was cooled down to the lowest temperature and left to equilibrate for 30 min. The microgels were added to the interface from solution with a concentration of 1 mg mL$^{-1}$ in deuterated chloroform or 50/50 vol% mixtures of water-propan-2-ol. Subsequently, the interface was compressed to $\approx 13$ mN m$^{-1}$ and the first measurement was conducted. At this surface pressure the average nearest neighbor distance between the microgels is $\approx 500$ nm as determined from AFM, see Supplementary Fig. 5. Afterwards the trough was tempered to the next temperature, left to equilibrate for 30 min, and subsequently, a measurement was conducted. This was repeated until 40 °C was reached. A feedback loop controlled and adjusted the surface pressure during the experiments. Surface pressures were measured with electric balances equipped with paper Wilhelmy plates.

In the literature, it is shown that the polymer fraction within ULC microgels in bulk is much lower than for cross-linked microgels[39,54,72]. As a consequence, their contrast is very low both in the bulk and at the interface, and long measurement times would be required to collect statistically reliable data. For this reason, only deuterated ULC microgel was measured at the interface. The substitution of 3 atoms of hydrogen with 3 atoms of deuterium improves the contrast of the ULC microgels when both ACMW and pure D$_2$O are used for the water phase.

**Analysis and model for neutron reflectometry data**. As mentioned above, SNR allowed us to determine the density profile of the microgel monolayer in situ along the z-direction, normal to the interface. The measured R(Q) profile can be linked to an in-plane averaged scattering length density (SLD) profile of the monolayer along the z-direction, $b(z)$, thus giving information of a statistically significant number of microgels.

Here, SNR data modeling was performed by minimizing the difference between the experimental and the calculated reflectivity profile using the Parratt's recursive formalism[82]. The calculated profiles were obtained under the assumption that the z-profile of the SLD can be decomposed in N-layers, with an error function connecting adjacent layers. Every layer was characterized by a constant scattering length density $b_i$, which depends on the volume fraction of polymer and solvent in this layer. Data analysis was performed using constraints between layer parameters (thickness, roughness, and degree of hydration or SLD) and simultaneous co-refinement of data sets at two contrasts (D$_2$O and ACMW) to reduce ambiguity in modeling with Motofit[83] in IGOR Pro (Wavemetrics). Thus, all parameters in Tables 2 and 3, except $b_i$, were co-refined for the two contrasts. The model was

fitted to the data using global minimization of a least-squares function $\chi^2$. In each $i$-layer, the SLD and the polymer fraction $x$ follows $b_i = x b_{pNIPAM} + (1 - x) b_{solvent}$, where $b_{pNIPAM}$ and $b_{solvent}$ are the theoretically calculated values. The polymer fraction distribution x(z) normal to the plane of the interface for each i-layer was calculated as the sum of two error functions as follows

$$x(z) = \frac{1}{2} x_i \left[ \text{erf} \left( \frac{z - d_i/2}{\sqrt{2}\sigma_i} \right) - \text{erf} \left( \frac{z + d_i/2}{\sqrt{2}\sigma_{i+1}} \right) \right], d_i < z < d_{i+1} \qquad (1)$$

where $d_i$ represents the length of the layer with scattering length density $b_i$. The roughness between two layers is given by $\sigma_i$. $\sigma_i$ denotes the roughness of a layer $i$ with the layer above $i - 1$. A similar model has been successfully used to fit NR-curves of pNIPAM nanogels[44,46].

For the regular microgels, $N$ was chosen equal four to satisfactorily fit the experimental curves. In contrast, good fits of the R(Q)s of a monolayer of ultra-low cross-linked microgels were obtained using three layers. In addition, to demonstrate that a Fresnel reflectivity calculation of a slab model that includes a Gaussian error function connecting the layers is valid even in our case, where the obtained roughness values are of the order of the layer thicknesses, an alternative model based on a continuous variation of the SLD profile was used. The SLD profiles were divided into many thin layers (1.5 Å), which sustain the same physical polymer fraction distribution. The results are compared in the Supplementary Information, Supplementary Fig. 10a–d. In particular, two sets of data (5 mol% D7 and ULC D3) were fitted with this alternative method (see Supplementary Information) yielding similar results and, therefore, validating the findings from the different slab-models used.

## Computer simulations

*Standard and ULC microgels modeling*. Individual microgels were obtained by self-assembling a binary mixture of patchy particles with valence two and four[71] mimicking the NIPAM monomers and the BIS cross-linkers, respectively. The assembly was carried out through the OXDNA simulation package[84]. Standard microgels were created from a total number of monomers $N \approx 42,000$ within a sphere with the radius $Z = 100\sigma_m$, where $\sigma_m$, the monomer diameter, is the unit of length in simulations. The cross-linkers, whose concentration was set to be 5% of the total number of monomers, experienced an additional designing force during the assembly so that they were more densely distributed in the center of the particle. The effect of this additional force has been extensively studied in a previous work[85]. For ultra-low-cross-linked (ULC) microgels, we used $N \approx 21,000$ and a sphere with $Z = 55.5\sigma_m$, as determined from the comparison of the form factors in bulk (Unpublished data – manuscript in preparation). In this case, the number of cross-linkers was set to 0.3% of the total number of monomers. In both standard and ULC microgels, the assembly was carried out until >99.9% of the possible bonds in the network were formed.

At this stage, reversible patchy interactions were made permanent by allowing the microgel beads to interact via the Kremer-Grest model[86], according to which all beads interact via the Weeks-Chandler-Anderson (WCA) potential:

$$V_{WCA}(r) = \begin{cases} 4\epsilon \left[ \left( \frac{\sigma_m}{r} \right)^{12} - \left( \frac{\sigma_m}{r} \right)^6 \right] + \epsilon & \text{if } r \leq 2^{\frac{1}{6}}\sigma_m \\ 0 & \text{otherwise.} \end{cases} \qquad (2)$$

where $\epsilon$ sets the energy scale and $r$ is the distance between two particles. Connected beads interacted also via the finitely extensible nonlinear elastic (FENE) potential,

$$V_{FENE}(r) = -\epsilon k_F R_0^2 \ln \left[ 1 - \left( \frac{r}{R_0 \sigma_m} \right)^2 \right] \text{ if } r < R_0 \sigma_m, \qquad (3)$$

with $k_F = 15$ which determines the stiffness of the bond and $R_0 = 1.5$ is the maximum bond distance.

To account for the responsivity of the microgel at different temperatures, monomers also interact via an additional potential

$$V_\alpha(r) = \begin{cases} -\epsilon\alpha & \text{if } r \leq 2^{1/6}\sigma_m \\ \frac{1}{2}\alpha\epsilon \left\{ \cos \left[ \gamma \left( \frac{r}{\sigma_m} \right)^2 + \beta \right] - 1 \right\} & \text{if } 2^{1/6}\sigma_m < r \leq R_0 \sigma_m \\ 0 & \text{if } r > R_0 \sigma_m \end{cases} \qquad (4)$$

with $\gamma = \pi \left( \frac{9}{4} - 2^{1/3} \right)^{-1}$ and $\beta = 2\pi - \frac{9}{4}\gamma$[87]. $V_\alpha$ introduces an effective attraction among polymer beads, modulated by the parameter $\alpha$, whose increase allows to mimic the collapse of the microgel observed at high temperatures.

*Behavior at the interface*. To investigate the behavior of a microgel adsorbed at an interface, we reproduced the effects of the surface tension by placing a microgel between two fluids. Such fluids were modeled with soft beads within the dissipative particle dynamics (DPD) framework[88,89]. The total interaction force among beads is $\vec{F}_{ij} = \vec{F}_{ij}^C + \vec{F}_{ij}^D + \vec{F}_{ij}^R$, where:

$$\vec{F}_{ij}^C = a_{ij} w(r_{ij}) \hat{r}_{ij} \qquad (5)$$

$$\vec{F}_{ij}^{D} = -\gamma w^2(r_{ij})(\vec{v}_{ij} \cdot \hat{r}_{ij})\hat{r}_{ij} \tag{6}$$

$$\vec{F}_{ij}^{R} = 2\gamma \frac{k_B T}{m} w(r_{ij})\frac{\theta}{\sqrt{\Delta t}}\hat{r}_{ij} \tag{7}$$

where $\vec{F}_{ij}^{C}$ is a conservative repulsive force, with $w(r_{ij}) = 1 - r_{ij}/r_c$ for $r_{ij} < r_c$ and 0 elsewhere, $\vec{F}_{ij}^{D}$ and $\vec{F}_{ij}^{R}$ are a dissipative and a random contribution of the DPD, respectively; $a_{ij}$ quantifies the repulsion between two particles, $\gamma = 2.0$ is a friction coefficient, $\theta$ is a Gaussian random variable with zero average and unit variance, and $\Delta t = 0.002$ is the integration time-step. Following previous works[10,37], we chose $a_{11} = a_{22} = 8.8$, $a_{12} = 31.1$, for the interactions between fluid 1 and fluid 2. Instead, for the monomer-solvent interactions we chose $a_{m1} = 4.5$ and $a_{m2} = 5.0$. In this way, we made fluid 1 the preferred phase for the microgel particle. The cut-off radius was always set to be $r_c = 1.9\sigma_m$ and the reduced solvent density $\rho_{DPD} = 4.5$. In this way, the total number of particles was about $2.6 \times 10^6$ for simulating standard microgels and $\approx 5.3 \times 10^6$ for ULC microgels. The reduced temperature of the system $T^*$ was fixed to 1 via the DPD thermostat. We note that by adjusting $V_a$ to reproduce the effect of temperature on the microgel, we did not alter the features of the interface, which remains defined by the DPD parameters listed above. Simulations were performed with the LAMMPS simulation package[90].

## Data availability

Raw data were generated at the Institute Laue-Langevin (ILL, Grenoble, France) using the Fluid Interfaces Grazing Angles Reflectometer (FIGARO). The NR raw data used in this study are available in the ILL Data Portal database under accession code 10.5291/ILL-DATA.9-11-1871[91] and 10.5291/ILL-DATA.EASY-462[92]. The raw data, associated data, and derived data supporting the results of this study have been deposited in the RADAR4Chem database under https://doi.org/10.22000/603[93] or are available from the corresponding author at the link http://hdl.handle.net/21.11102/b0e200f4-d196-44bd-874a-2f5f79d22527.

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

## Acknowledgements

The authors thank Yuri Gerelli for valuable discussions and Monia Burgnoni for synthesis of the deuterated microgels. The authors also thank Jerome J. Crassous and Andrea Ninarello for an ongoing collaboration on the modeling of ULC microgels in bulk. S.B., M.M.S., W.R., and A.S. acknowledge funding from the Deutsche Forschungsgemeinschaft within SFB 985 "Functional Microgels and Microgel Systems", projects A3 and B8. F.C. and E.Z. acknowledge financial support from the European Research Council (Consolidator Grant 681597, MIMIC). AM acknowledge the financial support received from the IKUR Strategy under the collaboration agreement between Ikerbasque Foundation and Materials Physics Center on behalf of the Department of Education of the Basque Government. This work is based upon NR experiments performed at the Institute Laue-Langevin (ILL, Grenoble, France) using the Fluid Interfaces Grazing Angles Reflectometer (FIGARO). This work is partially based on SANS experiments performed at the D11 instrument at the Institut Laue-Langevin (ILL), Grenoble, France and at the KWS-2 instrument operated by JCNS at the Heinz Maier-Leibnitz Zentrum (MLZ), Garching, Germany.

## Author contributions

W.R., A.S., E.Z., F.C., and S.B. designed this study. A.S., M.S., A.M., and S.B. performed the NR measurements. A.S., A.M., and S.B. designed the model for the NR data. A.M. and S.B. analyzed the NR data. S.B. synthesized and characterized the hydrogenated microgels. S.B. performed Langmuir-Blodgett and AFM measurements. S.B. analyzed the AFM data. F.C. performed the computer simulations. All authors participated in discussing the results, writing, finalizing, and revising the manuscript.

## Funding

## Competing interests

The authors declare no competing interests.
