## [Peer Review File · Nature Communications]

In-situ study of the impact of temperature and architecture on the interfacial structure of microgelsReviewers' Comments:

Reviewer #1:

Remarks to the Author:

The work by Bochenek and co-workers reports interesting neutron reflectometry (NR) results on smart polyacrylamide based microgels at the air water interface. To my knowledge at least for so-called ultra-low crosslinked (ULC) microgels this is the first time that such data are presented. Moreover, the experimental data are complimented by MD simulations which show very similar density profiles perpendicular to the interface as the NR experiments do. Hence, I believe that this work is important to better understand the behavior of microgels at interfaces in general which is relevant for a better understanding for their use in "smart" emulsions.

It is remarkable that the ULC microgels completely flatten at the interface. This is found in the experiment and independently in the simulations. The agreement between experiment and simulation shows the validity of the NR data analysis in very convincing way.

Hence, I think this is nice work which should be accepted after revision.

Unfortunately, the text is partly hard to read. This is e.g. due to the fact that figures are referenced much earlier than they appear in the text. Moreover, table 2 is not very helpful and hard to read even with glasses...

In fact the information of tab.2 is included plots of the polymer fraction perpendicular to the surface.

Hence, I suggest to include the thicknesses of the slabs used to model the reflectometry curves in these plots, maybe as vertical lines and to move the table to the SI.

Moreover, some sentences should be rephrased to enhance readability.

Reviewer #2:

Remarks to the Author:

Bochenek et al. report on the 'Interfacial Structure of Thermo-responsive Microgels' using neutron reflectometry and realistic in silico synthesized microgels via computer simulations. The study is very well written, coherent, and technically sound. The topic is also interesting and relevant, e.g. for microgels that act as stabilizers for emulsions that can be broken on demand, as discussed in the text. The authors compare the behavior of standard microgels (synthesized with 5% crosslinker) and ultra-low crosslinked microgels and find significant differences between both. While not perfect, the comparison between experiment and modeling is sound. Since the results from neutron reflectivity depend on model fitting, the comparison with realistic modeling is crucial in my view. The paper meets the standards of Nature Communication, and the text is suitable for publication.

Minor points: I would suggest the author's comment and discuss in more details the limitations and pitfalls of the fitting approach - how stable are the results for the density profiles e.g., shown in fig 2,4,5? More details are included in the SI, but some comments in the text for the general audience, without detailed knowledge about neutron reflectometry, would be helpful. In the 'Discussion'-section, the authors also claim that microscopy could not measure the protrusion of the microgels in the air. With the advent of novel superresolution techniques, however, detecting protrusions on the order of 30nm seems feasible, and thus, the authors should rephrase this statement.

Reviewer #3:

Remarks to the Author:

The authors of the paper "In-situ Study on the Interfacial Structure of Thermo-responsive Microgels: Impact of Temperature and Architecture" describe the properties of microgels at the air-water interface, in view of the responsiveness of certain microgels such as PNIPAM.

Neutron reflectometry and MD computer simulations have been employed to study the interfacial behaviour, especially the extend of the microgel into the air phase and possible contact angles of the air phase part of the microgel. Two different crosslink densities, a "conventional" and a rather low one,

have been investigated.

The fact that soft microgels deform stronger at interfaces has been observed by several groups, with AFM or neutron reflectometry for example for the solid-liquid interface, and also the picture at the air-water interface is not entirely new (e.g. Deshmukh, O. S., Maestro, A., Duits, M. H., van den Ende, D., Stuart, M. C., & Mugele, F. (2014). *Soft matter*, 10(36), 7045-7050.).

The new aspect here is to apply neutron reflectometry for obtaining the z-profile of the polymer density of ULC microgels across the interface on nanometer length scales, and combining this with MD simulations on the same length scale. It turns out that for the ULD microgels the MD simulations do not capture the details of interface adsorption (remaining microgel like particle in the water phase observed in NR, fully adsorbed flat polymer layer in MD).

In my understanding, the motivation for this study, a "breaking on demand" of emulsions, rather requires liquid-liquid interfaces and not liquid-air interfaces. This seems to be a bit far-fetched to me and would rather be the motivation for the paper of Schmidt et al (Langmuir 2020) of the same group. Generally, I think that this work is a very solid piece of research with interesting results which deserves publication after some clarifications. But I am not sure if Nature Comm. is the right place, the significance for this is in my opinion not plausibly explained. The relevance of the air-water interface needed to be pointed out in more detail, and also where this contribution goes much beyond the findings of refs 44 (Zielinska et al.) and 45 (Richardson et al.). Between an ULC microgel and a polymer chain is at a certain point not much of a difference, and the MD simulations do not capture the details of the experiment.

Some technical points which have to be addressed in my opinion in more detail:

- Why do the deuterated and "normal" PNIPAM microgels with the same cross-link density have a different swelling behaviour? (The swelling ration from hPNIPAM to dPNIPAM is about the same as from dPNIPAM to ULC).
- Fig.2: The polymer contents at the interface seems at first sight rather different for hPNIPAM and dPNIPAM(it should be the integral under the curves in a) and b), although the particles should be rather identical. Why?
- Fig.2: The MD simulations result in a significantly broader peak at $z=0$. Is the density in c) the same as the polymer fraction in a) and b)? The $\alpha=0$ simulation is then much less swollen than the "true" particle?
- The NR model contains several slabs and roughnesses for each slab. This makes it rather difficult to put everything together. Why not taking a many (say 100) slab model and getting rid of the roughness?
- What are the special conclusions of the d7-PNIPAM? It is not very obvious to me where the gain in information over the hPNIPAM is.

I. Reviewer #1

The work by Bochenek and co-workers reports interesting neutron reflectometry (NR) results on smart polyacrylamide based microgels at the air water interface. To my knowledge at least for so-called ultra-low crosslinked (ULC) microgels this is the first time that such data are presented. Moreover, the experimental data are complimented by MD simulations which show very similar density profiles perpendicular to the interface as the NR experiments do. Hence, I believe that this work is important to better understand the behavior of microgels at interfaces in general which is relevant for a better understanding for their use in “smart” emulsions. It is remarkable that the ULC microgels completely flatten at the interface. This is found in the experiment and independently in the simulations. The agreement between experiment and simulation shows the validity of the NR data analysis in very convincing way. Hence, I think this is nice work which should be accepted after revision.

Reply:

We thank the Reviewer for appreciating our work and especially the extent to which experiments and simulations complement each other.

Unfortunately, the text is partly hard to read. This is e.g. due to the fact that figures are referenced much earlier than they appear in the text. Moreover, table 2 is not very helpful and hard to read even with glasses... In fact the information of tab.2 is included plots of the polymer fraction perpendicular to the surface. Hence, I suggest to include the thicknesses of the slabs used to model the reflectometry curves in these plots, maybe as vertical lines and to move the table to the SI.

Moreover, some sentences should be rephrased to enhance readability.

Reply:

We followed the suggestion of the Reviewer regarding the tables in which now we do

Figure A1: (A) SLD profiles of 5 mol% cross-linked NIPAM microgels at different temperatures. Vertical lines indicate the thickness of the 4 layers at different temperatures.

not show the errors, allowing for a larger font size. The complete tables, including the respective uncertainty on the fitting parameters, can now be found in the Supplementary Information. Instead, by including the values of the slabs as vertical lines in the figures, unfortunately the readability is not improved (see Figure A1).

Regarding the position of the figures in the text, we believe that it will be fixed during the post-acceptance process (the journal indicated to put all the figures at the end of the manuscript). We also simplified our exposition in different parts of the manuscript.

II. Reviewer #2

Bochenek et al. report on the ‘Interfacial Structure of Thermo-responsive Microgels’ using neutron reflectometry and realistic *in silico* synthesized microgels via computer simulations. The study is very well written, coherent, and technically sound. The topic is also interesting and relevant, e.g. for microgels that act as stabilizers for emulsions that can be broken on demand, as discussed in the text. The authors compare the behavior of standard microgels (synthesized with 5% crosslinker) and ultra-low crosslinked microgels and find significant differences between both. While not perfect, the comparison between experiment and modeling is sound. Since the results from neutron reflectivity depend on model fitting, the comparison with realistic modeling is crucial in my view. The paper meets the standards of Nature Communication, and the text is suitable for publication.

Reply:

We are very glad for the Reviewer's appreciation of our work and for recommending it suitable for publication in *Nature Communications* after minor revision.

Minor points: I would suggest the author's comment and discuss in more details the limitations and pitfalls of the fitting approach - how stable are the results for the density profiles e.g., shown in fig 2,4,5? More details are included in the SI, but some comments in the text for the general audience, without detailed knowledge about neutron reflectometry, would be helpful.

Reply:

To test the robustness of the fits performed with our four- and three-slabs model, we analyzed some of the data sets using a model where more than 1000 layers with a thickness of 1.5 Å are used to produce a SLD distribution that leads to good fits of the reflectivity curves. We now show that the results of this model are the same as the one obtained in the original manuscript. As a general approach, from scattering to reflectivity, we always choose the model that has the less number of free parameters and lead to a good reproduction of the experimental data. This is done to minimize the limitation and pitfall, mentioned by the Reviewer, such as having the minimization process trapped in a local minima in the phase-space of the fitting parameters.

We have added the explicit reference to the results of the fit with a N-slabs model ($N > 1000$) in the manuscript in pages 5, 8, 9 and 14. Furthermore, we have added a section to the Supplementary Information where we report the comparison with the fits that use the N-slabs model. This comparison shows that this model leads to identical results than the fits performed using the four- (regular microgels) and the three-slabs (ULC) models.

In the 'Discussion'-section, the authors also claim that microscopy could not measure the protrusion of the microgels in the air. With the advent of novel superresolution techniques, however, detecting protrusions on the order of 30nm seems feasible, and thus, the authors should rephrase this statement.

Reply:

Super-resolution techniques are indeed powerful to investigate this kind of systems. However, we note that the resolution for super-resolved fluorescence microscopy (SRFM) on the z -direction, which would be required to detect the protrusion into air, is still limited to ≈ 60 nm [G. M. Conley et al. *Colloids and Surfaces A: Physicochem. Eng. Aspects* (2016) 499: 18-23]. Independently of this, to implement the blinking of the dye, the sample should be surrounded by the solvent. Another big challenge relates to the nature of the data in a SRFM experiment. Indeed, they consist of point clouds and the challenge is extract regions of interest that correspond to individual nanogel from a noisy

three dimensional set of experimental points. To overcome these technical challenges, at the moment there is a considerable effort devoted to develop new analysis-tools. For instance, recent approaches based on Voronoi tessellation [L. Andronov et al. Scientific Report (2016) 6: 24084] and model-based Bayesian clustering [P. Rubin-Delanchy et al. Nature Methods (2015) 12: 1072] have been proposed for detection of point clusters in super-resolution microscopy images. For all these reasons, we believe it is still very difficult to perform such kind of experiments and analysis.

We included these considerations on the SRFM in the Discussion.

III. Reviewer #3

The authors of the paper “In-situ Study on the Interfacial Structure of Thermo-responsive Microgels: Impact of Temperature and Architecture” describe the properties of microgels at the air-water interface, in view of the responsiveness of certain microgels such as PNIPAM. Neutron reflectometry and MD computer simulations have been employed to study the interfacial behaviour, especially the extend of the microgel into the air phase and possible contact angles of the air phase part of the microgel. Two different crosslink densities, a “conventional” and a rather low one, have been investigated. The fact that soft microgels deform stronger at interfaces has been observed by several groups, with AFM or neutron reflectometry for example for the solid-liquid interface, and also the picture at the air-water interface is not entirely new (e.g. Deshmukh, O. S., Maestro, A., Duits, M. H., van den Ende, D., Stuart, M. C., & Mugele, F. (2014). *Soft Matter*, 10(36), 7045-7050.). The new aspect here is to apply neutron reflectometry for obtaining the z-profile of the polymer density of ULC microgels across the interface on nanometer length scales, and combining this with MD simulations on the same length scale. It turns out that for the ULC microgels the MD simulations do not capture the details of interface adsorption (remaining microgel like particle in the water phase observed in NR, fully adsorbed flat polymer layer in MD).

Reply:

We respectfully disagree with the Reviewer when he/she says that our computer simulations “do not capture the details of interface adsorption”. As indicated in the text, the comparison between experiments and simulations for ULC microgels is consistent in several aspects, considering that (i) the contact angle between polymer and interfacial plane is zero, (ii) the thick layer of polymer sitting onto the interface is comparable to the one of the regular nanogels and (iii) they show no protrusion in the oil phase, with a slight preference for the water phase. The main difference is that in the simulations

there is no small protrusion of the ULC microgels into the water phase. It is to be expected that this is generated by a series of dangling chains, provided the absence of crosslinkers during the synthesis of this type of microgel. The reason why this detail cannot be captured in the simulations is related to the size of the microgel, for which we expect that for having substantial differences the number of monomers should be at least an order of magnitude higher than the one simulated here (a size effect comparison for standard microgels has been previously carried out in [Ninarello et al. *Macromolecules* (2019)]). Considering the presence of explicit solvent to simulate interfacial effects, the low crosslinking content and the high tendency of the microgel to increase its size when confined, such kind of simulations are currently unfeasible. For these reasons, we believe that the comparison between simulations and experiments is sound for both standard and ULC microgels, with limitations due to purely technical reasons already illustrated and now further emphasized in the text.

These considerations were already included and discussed in the previous manuscripts. We now highlight them in pages 10 and 11 for more clarity.

In my understanding, the motivation for this study, a “breaking on demand” of emulsions, rather requires liquid-liquid interfaces and not liquid-air interfaces. This seems to be a bit far-fetched to me and would rather be the motivation for the paper of Schmidt et al (Langmuir 2020) of the same group. Generally, I think that this work is a very solid piece of research with interesting results which deserves publication after some clarifications. But I am not sure if Nature Comm. is the right place, the significance for this is in my opinion not plausibly explained. The relevance of the air-water interface needed to be pointed out in more detail, and also where this contribution goes much beyond the findings of refs 44 (Zielinska et al.) and 45 (Richardson et al.). Between an ULC microgel and a polymer chain is at a certain point not much of a difference, and the MD simulations do not capture the details of the experiment.

Reply:

We thank the Reviewer for the comment “Between an ULC microgel and a polymer chain is at a certain point not much of a difference” as this indicates that we need to discuss that more clearly. We, however, respectfully disagree with the reviewer’s statement. Already in bulk, ULC microgels and linear or branched polymer chains have a different behavior, with the former forming colloidal crystals [A. Scotti et al. *Physical Review E* (2020) 102: 052602]. Therefore, the presence of cross-linkers, albeit low, does make a significant difference. Concerning the behavior of ULC microgels at interfaces: We showed that ULC microgels can tune their behavior between the one of polymer and the one of colloidal particle depending on the compression of the monolayer [A. Scotti et al. *Nature Communications* (2019) 10: 1418]. In our recent *Angewandte Chemie*

[M. F. Schulte et al. *Angewandte Chemie International Edition* (2021) 60: 2280-2287], we also showed that the ULC microgels share a great deal with colloidal particles. In addition to our contributions, recently, other groups studied ULC nanogels and showed how their properties lies in between the one of polymer and colloidal particles, but for sure they are much different with respect to simple polymer chains [e.g. A. C. Brown et al. *Nature Materials* (2014) 13: 1108-1114, and M. R. Islam et al. *Macromolecular Rapid Communications* (2021): 42, 2100372].

We added an explicit discussion of the recent literature that shows that the ULC microgels are fundamentally distinct from polymer chains in page 3.

Regarding the cited works of Campbell and co-workers, we note that they have been conducted either on linear-pNIPAM or on nanogels with diameters of the orders of ≈ 40 nm. The latter have a mesh-size comparable to their radius and, therefore, one can expect a significant difference in the stretching of these particles at the interface compared to microgels with radii larger than ≈ 100 nm. Indeed, as we mentioned already in the conclusions the behavior of the ULC *is consistent with the experiments of Richardson and co-workers that used neutron reflectivity to probe linear pNIPAM solutions and nanogels with a mesh-size comparable to their dimensions and, therefore, highly stretchable at the interfaces.* The novelty of our work is to compare microgels with a clear different internal architecture at the interface and study the effect of temperature on their radial distribution in the z -direction.

Furthermore, *we added in the discussion, on page 12, that “recent literature has also shown that the substitution between air and alkanes, such as decane, only slightly changes the stretching of the microgels at the interface [S. Bochenek, A. Scotti, and W. Richtering, *Soft Matter* (2021) 17: 976]. This is due to high interfacial tension of the two systems and the insolubility of the microgels in the alkane/oil. However, at a lower interfacial tension, a greater reduction in the spreading of the microgels is observed [J. Vialetto et al. *J. of Colloid and Interface Science* (2022) 608: 2584-2592]. Therefore, we expect that our results on the protrusion of the microgels into the hydrophobic phase and the observed difference between ULC and standard microgels at an alkane/(oil)-water interface will not change qualitatively”.*

Some technical points which have to be addressed in my opinion in more detail:

- Why do the deuterated and “normal” PNIPAM microgels with the same cross-link density have a different swelling behaviour? (The swelling ration from hPNIPAM to dPNIPAM is about the same as from dPNIPAM to ULC).

Reply:

The different swelling behavior of the deuterated and hydrogenated microgels is an effect of the deuteration. As reported in the literature, both selective deuteration and the use of deuterated solvents leads to a shift of the volume phase transition temperature of pNIPAM [H. Shirota et al. Chemical Physics (1999) 242:115-121 and H. Shirota et al. The Journal of Physical Chemistry B 1999, 103, 10400-10408]. However, at 20 and 40 °C, both the deuterated and hydrogenated nanogels are in the swollen and completely collapsed state, respectively. This means that the swollen and collapsed state for the hydrogenated and deuterated microgels we adsorb at the interface are qualitatively comparable even if the swelling behavior is slightly different. In a recent paper, also Hellweg and co-workers observed this behavior and tried to rationalize this shift in terms of a change in the strength of the hydrogen bonds between the partially deuterated monomer and the solvent [M. Cors et al. Polymers (2019): 11, 620]. Finally, Buratti et al. also observed a change in the swelling behavior upon deuteration [E. Buratti et al. J. Mol. Liq. 355, 118924 (2022)].

We explicitly mention the effects of deuteration in Page 3 and we added the references mentioned above.

- Fig.2: The polymer contents at the interface seems at first sight rather different for hPNIPAM and dPNIPAM (it should be the integral under the curves in a) and b), although the particles should be rather identical. Why?

Reply:

The two particles are slightly different since they come from two different batches and due to use of a deuterated monomer. This might change slightly the reaction affecting for instance the monomer incorporation and the ratio between length of the shell and radius of the core. Following Romeo et al. [J. Chem. Phys. (2012) 136: 124905], we estimate the microgel mass and molecular weight (M_w), combining viscosimetry measurements and dynamic light scattering. The regular deuterated microgels have a mass of $6.3 \pm 0.6 \cdot 10^{-19}$ kg ($M_w = 3.8 \pm 0.4 \cdot 10^8$ gmol $^{-1}$), while the regular hydrogenated nanogels have a mass of $7.7 \pm 0.7 \cdot 10^{-19}$ kg ($M_w = 4.6 \pm 0.4 \cdot 10^8$ gmol $^{-1}$). This means that, even if the regular microgels have a similar core-fuzzy shell architecture, the deuterated nanogels have less polymer to arrange at the interface. This is the reason for the quantitative difference between the polymer content at the interface. However, we note that the amount of polymer on the interfacial plane is remarkably identical for both microgels. The trend is also confirmed by the deuterated ULC microgels that have the smallest mass, $4.6 \pm 4 \cdot 10^{-19}$ kg ($M_w = 2.7 \pm 0.3 \cdot 10^8$ gmol $^{-1}$) showing indeed the lowest amount of polymer. Even in this case, the amount of polymer sitting at the interface is the same as for the regular hydrogenated and deuterated nanogels. This tells us that the amount of polymer in the water phase depends on the mass and structure of the adsorbed nanogels, while the amount of polymer sitting onto the interface is almost independent on these properties and the polymer network just try to minimize as much

as possible the interfacial tension.

We added an extract of these considerations in Pages 7 and 8.

- Fig.2: The MD simulations result in a significantly broader peak at $z=0$. Is the density in c) the same as the polymer fraction in a) and b)? The $\alpha=0$ simulation is then much less swollen than the "true" particle?

Reply:

The information that can be extracted from the comparison between the experimental and numerical curves concerns the three regions that are found once the microgel is absorbed at the interface, as described in the manuscript. In particular, in both experiments and MD simulations, we find a slight protrusion into the air due to the presence of the core, a high density intermediate region where the polymer network is absorbed between the two fluids, and the protrusion into the aqueous phase. A quantitative comparison between simulations and experiments would require, once again, a simulated microgel with a number of monomers of the order of 10^6 , such that the single monomer corresponds to the actual Kuhn length. Since this is not yet achievable, the density profiles are reported in σ units, which is the unit of length used in the simulations. This implies that a direct comparison, for instance related to the measure of the peak thickness at $z = 0$, mentioned by the Reviewer, is not possible at the moment. Nevertheless, the qualitative agreement on the particle morphology and the trend as a function of temperature are found to be consistent and sound.

These considerations have been further underlined in the manuscript, at pages 10 and 11.

- The NR model contains several slabs and roughnesses for each slab. This makes it rather difficult to put everything together. Why not taking a many (say 100) slab model and getting rid of the roughness?

Reply:

We would like to thank the Reviewer for having raised this point, which was also evoked by Reviewer #2. Please refer to the response given to Reviewer #2's comment. Following both Reviewers' suggestion, we performed the fits of some set of data using a model that consists of $N > 1000$ slabs with a fixed thickness of 1.5 \AA . The resulting SLD distributions are the same as obtained by four- and three-slabs models, respectively.

We included a section in the Supplementary Information where we show the results and comparison of two data-sets fitted with the different models. We also mention in the manuscript, in pages 5, 8, 9 and 14, that the fits with the N -slabs model lead to the same SLD distributions.

- What are the special conclusions of the d7-PNIPAM? It is not very obvious to me where the gain in information over the hPNIPAM is.

Reply:

The aim of using the d7-pNIPAM nanogels was to improve the contrast between the microgels and the background. This allowed us to perform air-contrast matched water (ACMW) measurements in a reasonable amount of time, approximately 3-4 hours. As mentioned above, even if there are few quantitative differences between the crosslinked nanogels, the overall architecture and protrusion in the air phase is quite consistent. This ensures that if any additional cross-linker agent is used during the synthesis, a small but not negligible contact angle and some protrusion into the air will be observed.

We added an explicit reference to the improvement of contrast due to selective deuteration in page 3.

Reviewers' Comments:

Reviewer #2:

Remarks to the Author:

The authors have addressed my relatively minor concerns satisfactorily. I recommend the publication of this work in Nature Communications.

Reviewer #3:

Remarks to the Author:

The authors addressed the questions I had in an adequate manner. Therefore I can support publication of the manuscript, although I still have some difficulties understanding the bigger and surprising benefit for emulsion applications. Technically it is a very sound and thorough study.